# Identification of pathogenic *Leptospira* species and serovars in New Zealand using metabarcoding

**David A. Wilkinson** [1,2,3]*, **Matthew Edwards**[3], **Jackie Benschop** [3], **Shahista Nisa**[3]

**1** New Zealand Food Safety, Science & Research Centre, Massey University, Palmerston North, New Zealand, **2** UMR Processus Infectieux en Milieu Insulaire Tropical (PIMIT) INSERM 1187, CNRS 9192, IRD 249, Université de La Réunion, Sainte-Clotilde, La Réunion, France, **3** Molecular Epidemiology and Public Health Laboratory, School of Veterinary Sciences, Massey University, Palmerston North, New Zealand

* dwilkin799@gmail.com

**Data Availability Statement:** Draft genomes of New Zealand isolates are deposited in the NCBI database under Bioproject number PRJNA666522.

**Funding:** This project was funded by the Hawke's Bay Medical Research Foundation and the

## Abstract

Leptospirosis is a zoonotic disease of global importance. The breadth of *Leptospira* diversity associated with both human and animal disease poses major logistical challenges to the use of classical diagnostic techniques, and increasingly molecular diagnostic tools are used for their detection. In New Zealand, this has resulted in an increase in positive cases reported nationally that have not been attributed to the infecting serovar or genomospecies. In this study, we used data from all pathogenic *Leptospira* genomes to identify a partial region of the *glmU* gene as a suitable locus for the discrimination of the infecting species and serovars of New Zealand-endemic *Leptospira*. This method can be used in culture and culture-independent scenarios making it flexible for diagnostics in humans, animals, and environmental samples. We explored the use of this locus as a molecular barcoding tool via the Oxford Nanopore Technology (ONT) sequencing platform MinION. Sequences obtained by this method allowed specific identification of *Leptospira* species in mixed and enriched environmental cultures, however read error inherent in the MinION sequencing system reduced the accuracy of strain/variant identification. Using this approach to characterise *Leptospira* in enriched environmental cultures, we detected the likely presence of *Leptospira* genomospecies that have not been reported in New Zealand to date. This included a strain of *L. borgpetersenii* that has recently been identified in dairy cattle and sequences similar to those of *L. mayottensis*. *L. tipperaryensis*, *L. dzianensis* and *L. alstonii*.

## Introduction

Leptospirosis is a neglected zoonotic disease with a worldwide distribution [1]. It is estimated to cause 1.03 million cases and 58.9 thousand deaths annually, however, the true global burden of the disease is likely under-estimated [2]. Infection commonly results in undifferentiated febrile illness, which is often misdiagnosed, diagnostic tests are difficult to perform and there are limitations with current techniques [3].

Leptospirosis is caused by bacteria of the genus *Leptospira* [4]. This highly diverse genus has traditionally been divided into three major lineages: pathogenic, intermediate and

Palmerston North Medical Research Foundation. Our research into leptospirosis in New Zealand is additionally supported by the New Zealand Health Research Council.

**Competing interests:** The authors have declared that no competing interests exist.

saprophytic. However, recent isolation of new strains has led to a reclassification of the lineages. The genus is now divided into 2 major clades: pathogenic (P) and saprophytic (S). Each of these clades are subdivided into 2 subclades (P1, P2, S1 and S2); where P1 is the original pathogenic lineage that is virulent, P2 is the intermediate low-virulence clade, S1 is the original saprophytic lineage and S2 is the new saprophytic clade [5]. Pathogenic species primarily infect mammals (land and sea) and have been found in birds, amphibians, reptiles and fish [6]. *Leptospira* colonise the kidneys and are excreted into the environment through animal urine, where they may persist for extended periods while retaining their virulence [7]. Disease is transmitted by contact of mucosal membranes with the infected animals' urine or with the contaminated environment. Due to the multipartite nature of *Leptospira* reservoirs, leptospirosis surveillance and control require an interdisciplinary One Health approach to identify pathogens in humans, animals and the environment which can be useful for outbreak investigation [8,9].

The gold standard for leptospirosis diagnosis in humans and animals is a serological assay called the microscopic agglutination test (MAT) [4]. MAT requires an accurate knowledge of locally circulating serovars with regular surveillance to maintain a complete panel of antigens for relevant serological tests [3]. Some countries have a large number of circulating serovars making serological assays unfeasible and expensive, and testing is often limited to a small panel of reference isolates [10]. In addition, interpretation of results can be subjective requiring expert knowledge given known cross-reactivity patterns between serovars, and delayed or absent immune responses with antibodies taking up to two weeks to reach detectable levels post-symptom onset [10]. The classical methods for identifying *Leptospira* are visualization by dark field microscopy and culture [11]. However, dark field microscopy requires at least 10 *Leptopspira*/mL for visualization with a reasonable chance for false negative diagnosis [12]. Culture methods are laborious as *Leptospira* are fastidious bacteria that are often difficult to cultivate and it can take up to 13 weeks to grow in very specific culture conditions [4]. Furthermore, the cross-agglutinin absorption test (CAAT) that determines serovar is difficult, only performed by a limited number of laboratories around the world and, it cannot be used to distinguish pathogenic and saprophytic *Leptospira* [13].

To overcome the caveat of serological assays like MAT and CAAT, in more recent years, molecular techniques have been applied to *Leptospira* typing and detection. Molecular methods are often preferable to their serological alternatives in clinical settings due to their rapidity and increased sensitivity during early-stage infection [14]. Several genetic loci have been identified that are only present in the subclade P1 and these are often used for molecular diagnosis [15]. However, genetic and serological classification systems show poor correlation between the 64 genomospecies and more than 300 serovars known worldwide. One species can have multiple serovars or one serovar can be from multiple species [6]. Serological data are often reported without genetic data, or vice-versa, which limits the efficacy of public health surveillance and investigations [16].

Serosurveillance has played an important role in the control of leptospirosis in New Zealand. In the 1970's, there was a high incidence of leptospirosis due to serovars Hardjo and Pomona and, dairy cattle and pigs were found to be the maintenance hosts for these two serovars [17]. By the 1980's, widespread dairy cattle [18] and pig vaccination [19] against serovars Hardjo and Pomona was implemented as an intervention strategy to control disease in humans. Since the introduction of these intervention measures, leptospirosis incidence in New Zealand has continued to decline over the years until 2015 [20,21]. In the last 3 years, incidence has been steadily increasing. However, 30% of cases are reported as a positive or negative for *Leptospira*, with no serological data associated with them due to the recent implementation of DNA testing for diagnosis [22]. This can have serious implications for surveillance and

mitigation strategies as different serovars have different reservoir distributions [23] and the absence of infecting-lineage data means that likely sources of infection cannot be accurately attributed.

In this study, we linked the genetic and serological data of the major pathogenic species and serovars from subclade P1 of *Leptospira* circulating in New Zealand to identify a single locus capable of distinguishing pathogenic genomospecies endemic to New Zealand. The specific aim of this study was to develop a single, sensitive, and specific PCR system that could be used to address the shortcomings of current assays for *Leptospira* detection and diagnosis. Furthermore, we tested this PCR system for metabarcoding to identify individual *Leptospira* lineages from environmental or mixed samples. We sequenced PCR products from mixed, enriched environmental cultures using the Oxford Nanopore Technologies (ONT) sequencing platform MinION. The established PCR protocol showed high sensitivity, and sequencing results were specific at the species-level. In future, this assay could be applied to human and animal diagnostics as well as environmental sample characterisation allowing the standardization of typing methods for leptospirosis diagnostics in New Zealand. Furthermore, our results suggest that the full diversity of pathogenic *Leptospira* circulating in New Zealand is yet to be fully described and that further work will be needed to assess the risk of human infection from the newly identified *Leptospira* lineages.

## Methods

### Bacterial strains and growth conditions

*Leptospira* strains used in this study for metabarcoding controls are listed in S1 Table. All strains were cultured in liquid Ellinghausen-McCullough-Johnson-Harris (EMJH) medium supplemented with 0.3 μg/mL copper sulfate, 4 μg/mL zinc sulfate, 50 μg/mL ferrous sulfate, 15 μg/mL calcium chloride, 15 μg/mL magnesium chloride, 2 μg/mL cyanocobalamin, 1.25 mg/mL tween 80, 200 μg/mL sodium pyruvate, 100 μg/mL glycerol, 100 μg/mL sodium acetate, 10 g/L bovine serum albumin and 100 μg/mL 5' fluorouracil at pH 7.4. Cultures were monitored using dark field microscopy and maintained at 28˚C with shaking at 150 rpm.

### Environmental cultures

Environmental water samples were collected from a coastal dairy farm and a pine forest bordering the dairy farm in New Zealand. Permission to sample was granted by the land owner. Water was collected from puddles, ponds, ditches, animal troughs and creeks. A total of 24 samples were collected, 12 from the farm and 12 from the forest. 50 mL of water was collected per sample and centrifuged at 8 000 × g for 10 minutes. The supernatant was removed, leaving behind 500 μL of water. The pellet was resuspended in the remaining 500 μL of water and 200 μL of the concentrated water sample was inoculated into 5 mL EMJH media supplemented with STAF antibiotics (40 μg/mL sulfamethoxazole, 20 μg/mL trimethoprim, 5 μg/mL amphotericin B and 400 μg/mL fosfomycin) [24] to generate enriched cultures. Environmental enrichments were monitored for 13 weeks using dark field microscopy and maintained at 28˚C with shaking at 150 rpm. Contamination was eliminated by filtering cultures through 0.2 μm filters. Enriched cultures were pelleted for gDNA extraction and used in MinION sequencing.

### DNA extraction and quantification

Bacterial DNA was extracted using the QIAamp DNA Mini Kit (QIAGEN) and quantified using Qubit dsDNA HS Assay Kit (Invitrogen) as per manufacturer's instructions. An average

genome size of 4.70 Mb was used for *Leptospira interrogans* and 3.89 Mb was used for *Leptospira borgpetersenii* to determine the genomic equivalent per microliter of DNA for each species [25]. DNA from 2 species and 6 serovars (S1 Table) were serially diluted 10-fold from 200 000 to 0.02 genomic equivalent and used to determine the limit of detection (LOD). All 6 DNA preparations were also diluted to 5 ng/μL, pooled in 3 different ratios to create mixed mock communities (MMC) and used as templates for library preparation and MinION sequencing.

## Primer design

Genomic regions to identify pathogenic *Leptospira* are well defined in literature. Primer design took into consideration 88 published primers targeting 14 genes used for the detection of pathogenic *Leptospira* (S2 Table). Primers were assessed for mismatches and degeneracy against 598 genomes (495 published in NCBI database and 103 draft genomes of New Zealand isolates). Primers with mismatches were excluded from the analysis. The remaining primers were assessed for their ability to distinguish the 6 endemic serovars belonging to 2 species and 5 serogroups in New Zealand (S1 Table). The only loci with allelic variation for all 6 serovars was *glmU*. The *glmU* primers used in the MLST_2 scheme [26] were assessed for their ability to amplify specific products from DNA extracted from pure culture and from cattle urine [18]. The *glmU* MLST_2 primers amplified non-specific products. Other primers were designed for the *glmU* region and glmU_DW_F 5'-CCCGTATGAAAACGGATCAGCC-3' and glmU_DW_R 5'-ATTCTCCCTGAGCGTTTTGATTTC-3') primers were found to amplify specific product (S1 Fig). GlmU_DW amplicons were verified with Sanger sequencing.

## Sequence database and phylogeny

The locus amplified by the designed primers was extracted *in silico* from 598 available *Leptospira* genomes and aligned using MAFFT [27]. Unique sequences within the alignment were retained and assigned consecutive allele numbers. This alignment was used as the reference database to assign sequences to taxa in the downstream analyses. Reference sequences from the 6 relevant New Zealand genomospecies were included in the data set, and the 1:1 correspondence between species and allele sequence was verified for each genomospecies. In addition, a unique sequence obtained from the application of the *glmU* PCR to cattle urine that were positive to *Leptospira* [18], a *L. borgpetersenii* strain (informally referred to as *Pacifica*), was also included in the alignment. The phylogeny of the resulting allele sequence alignment was inferred using the Maximum Likelihood method based on the General Time Reversible model [28]. Initial trees for the heuristic search were obtained automatically by applying Neighbor-Join and BioNJ algorithms to a matrix of pairwise distances estimated using the Maximum Composite Likelihood (MCL) approach, and then selecting the topology with superior log likelihood value. All positions containing gaps and missing data were eliminated. There was a total of 523 base positions in the final dataset. Evolutionary analyses were conducted in MEGAX (v.10.1.6) [29]. For specific strain comparisons, single nucleotide polymorphisms were predicted in pairwise comparisons using SNIPPY v.4.6.0 (available at https://github.com/tseemann/snippy).

## PCRs

All primers used in this study (n = 10) are identified in S2 Table with an asterisk. TaqMan qPCR assays were performed with ToughMix (Quantabio) using *lipL32* primers and conditions as published [29]. Conventional PCR assays were performed with FIREPol master mix (Solis BioDyne) using primers specific for the *glmU* and *gyrB* loci [30]. Conventional PCR reaction conditions consisted of an initial denaturation step at 95°C for 5 minutes, followed by

5 cycles of 95°C for 30 seconds, annealing at 50°C for 30 seconds, extension at 72°C for 1 minute and another 50 cycles of 95°C for 30 seconds, annealing at 55°C for 30 seconds, extension at 72°C for 1 minute and a final extension at 72°C for 7 minutes. Primer concentrations in all conventional PCRs were 400 nM. All LOD PCRs were run in triplicate and control reactions without template were included in each assay.

## MinION sequencing and analysis

MinION libraries were prepared using a 1 step 4-primer PCR with GDW_long_F 5'-TTTCTGTTGGTGCTGATATTGCCCCGTATGAAAACGGATCAGCC-3' and GDW_long_R 5'-ACTTGCCTGTCGCTCTATCTTCATTCTCCCTGAGCGTTTTGATTTC-3' primers. The GDW_long primers contain the glmU_DW primer sequence with a 5' adapter tag. Library preparation contained 30 ng template, 50 nM of each GDW_long primers, 0.75 μL of barcode primers (SQK-LWB001 sequencing kit) and 1x LongAmp Hot Start Taq (New England Biolabs) in a final volume of 25 μL. The amplification protocol consisted of a denaturation step at 94°C for 1 minute, followed by 5 cycles of 94°C for 30 seconds, 55°C for 30 seconds, 65°C for 1 minute and another 30 cycles of 94°C for 30 seconds, 62°C for 30 seconds, 65°C for 1 minute and a final extension at 65°C for 5 minutes. The first 5 cycles were specific for the GDW_long primers and the subsequent 30 cycles were specific for the ONT adaptor sequences. PCR products were purified using AMPure XP beads (Beckman Coulter) as per manufacturer's instruction, quantified using Qubit and PCR with different barcodes were pooled in equal ratios. A concentration of 350 fmol to 1500 fmol of pooled DNA was used for MinION sequencing using a FLO-MIN106 Flow cell and the MinKNOW software for 2 hours and at a voltage of 180 (Oxford Nanopore Technologies).

Basecalling and demultiplexing was performed in Guppy V4.0.11, using the–require_-both_strands criterium to ensure minimal crosstalk between barcodes. Sequences were assigned to alleles based on best-hit BLAST scores using custom scripts. Where more than one reference sequence showed identical levels of homology to the queried sequence, the sequence assignment was deemed ambiguous, and its count was shared between the matching alleles.

# Results

## Primer design and testing

The *glmU* locus was identified as a suitable allele to distinguish species belonging to subclade P1 endemic to New Zealand. Optimised primers (glmU_DW_F and glmU_DW_R) that are fully conserved and without degeneracy against all subclade P1 species of *Leptospira* were used to amplify specific products (S1 Fig). The limit of detection of the glmU_DW PCR was comparable to the widely used *lipL32* Taqman qPCR and the *gyrB* conventional PCR (Table 1).

The glmU_DW primers were tested on gDNA extracted from the urine of cattle that were shedding *Leptospira* that was identified with the *gyrB* primers [18]. This revealed an allelic variation of *L. borgpetersenii* found in over 50% of *gyrB* positive urines. This strain (informally called *Pacifica*) is believed to be endemic in New Zealand, however, no isolate has been cultured to date. Pairwise alignment of the *glmU* region from 598 genomes identified 104 different alleles, including *Pacifica* (Fig 1).

All *Leptospira* genomes from New Zealand isolates belonged to serovars that had been verified using in-house MATs, whereas serovar data were not available for all international genome sequences. Therefore, New Zealand-specific *glmU* alleles were attributed to their corresponding serovars based on the following information: Allele 1 (*L. borgpetersenii* sv. Hardjo type Bovis) was extracted from 35 genomes, of which 10 isolates were verified by MAT, allele 2 (*L. interrogans* sv. Pomona) was extracted from 65 genomes, 9 of which were verified by MAT,

**Table 1. Limit of detection of PCR systems tested against *Leptospira* strains endemic to New Zealand.**

| Genomopecies | Serovar | Taqman qPCR | cPCR | |
|---|---|---|---|---|
| | | LipL32 | GyrB | GlmU_DW |
| *L. borgpetersenii* | Ballum | 2 | 20 | 2 |
| | Balcanica | 2 | 2 | 20 |
| | Hardjo type Bovis | 20 | 2 | 20 |
| | Tarassovi | 20 | 0.2 | 20 |
| *L. interrogans* | Copenhageni | 20 | 2 | 2 |
| | Pomona | 2 | 20 | 20 |

Limits of detection are expressed as the estimated number of copies of *Leptospira* genomes in the least concentrated sample of a serial dilution that gave a positive PCR result.

allele 3 (*L. borgpetersenii* sv. Ballum) was extracted from 89 genomes, 54 of which were verified by MAT, allele 4 (*L. borgpetersenii* sv. Balcanica-Burgas) was extracted from 2 genomes, 1 of which was verified by MAT, allele 5 (*L. borgpetersenii* sv. Balcancia-NZ) was extracted from 4 genomes, all of which were verified by MAT, allele 6 (*L. interrogans* sv. Copenhageni) was extracted from 174 genomes, 2 of which were verified by MAT, allele 66 (*L. borgpetersenii* sv. Tarassovi) was extracted from 9 genomes, 6 of which were verified by MAT and allele 104 (*L. borgpetersenii* strain Pacifica) was extracted from 61 urine/kidney samples, none of which was verified with MATs as there are no isolate available to date, however, cattle shedding this strain have been shown to have high titres against serovar Tarassovi [18].

Allele 3, which is used to identify *L. borgpetersenii* sv. Ballum in New Zealand is identical to *L. borgpetersenii* sv. Arborea thus serovars Arborea and Ballum cannot be distinguished using the *glmU* locus. Serovars Arborea and Ballum also cannot be distinguished by MATs because of a high level of cross-reactivity between them. The homologous reaction of live leptopsires of serovar Ballum against Ballum antiserum produces a titre of 1:10240 while the heterologous reaction of live leptospires serovar Arborea against Ballum antiserum and produces a titre of 1:1280 (*Leptospira* Reference Centre, Academic Medical Centre, Netherlands). Whole genome sequence analysis of *L. borgpetersenii* sv. Ballum strain mus127 and *L. borgpetersenii* sv. Arborea strain Arborea shows a total of 105 single nucleotide polymorphisms between the two serovars. While there has been some serological evidence of Arborea in New Zealand deer [31], to date, Arborea has not been isolated in New Zealand.

Allele 5, which is used to identify *L. borgpetersenii* sv. Balcanica (NZ), appears to be specific to New Zealand and to date has been identified in New Zealand possums and deer and is different from the commonly known Balcanica Burgas strain at the *glmU* locus.

Allele 6, which is used to identify *L. interrogans* sv. Copenhageni in New Zealand is identical to *L. interrogans* sv. Hardjo type Prajitno, however, *L. interrogans* sv. Hardjo type Prajitno is not known to exist in New Zealand [32].

## Metabarcoding

The glmU_DW primers were tagged with the Oxford Nanopore universal tags (GDW_long primers) and used for barcoding 6 strains representative of the endemic species and serovars in New Zealand. Nanopore sequencing results of the individually barcoded strains showed that 100% of the sequences were assigned correctly at the species level for all control strains except for Hardjo type Bovis control from species *L. borgpetersenii* which accurately assigned 84% of the sequences (Fig 2). In addition, approximately 95% of sequences were assigned correctly at the allele level for all control strains except for *L. interrogans* sv. Copenhageni, which

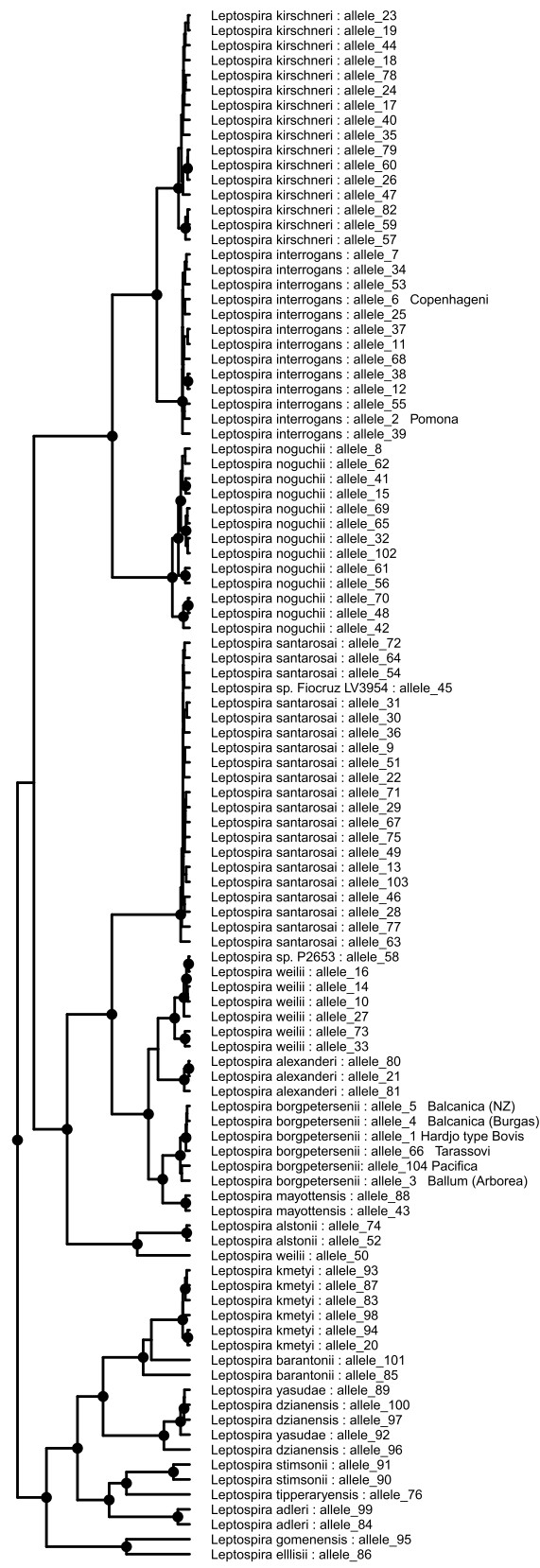

**Fig 1. Phylogenetic tree based on the *glmU* sequences of *Leptospira*.** The tree is drawn to scale, with branch lengths measured in the number of substitutions per site. Alleles associated with serovar information have been cross-verified using in-house MATs.

only had approximately 50% of the sequences classified correctly while the other 50% mapped to non-endemic *L. interrogans* alleles (Fig 2). Mapping the individually barcoded serovars to all the alleles, included in the phylogenetic tree shown in Fig 1, showed some misclassification of alleles between members of the same species, which can be attributed to the error rate of the MinION sequencing platform (Fig 3).

This sequencing method was applied to mock communities to validate the metabarcoding approach for *Leptospira* for species and serovar identification in samples containing a mixture of species. Three mock communities with varying concentrations of DNA per serovar were created. Results from triplicate experiments consistently estimated the proportions of DNA originating from each control strain, although assignment accuracy varied between control strains. As was observed for the individually barcoded strains, metabarcoding showed the least accurate assignment rates for control strains *L. borgpetersenii* sv. Hardjo type Bovis and *L. interrogans* sv. Copenhagenii (Fig 4) and, the *L. interrogans* sv. Copenhagenii mapped to many non-endemic *L. interrogans* alleles. This misassignment appeared to be dependent on the concentration of Copenhageni DNA (Fig 3). Interestingly, near-zero misassignment was observed to allele 104 (Pacifica) in both single isolate and mock community control experiments. This is due to the fact that the genetic signature of *Pacifica* is significantly divergent from that of other isolates of *L. borgpetersenii* in the database, and thus random sequencing errors are highly unlikely to result in misassignment to this allele (Fig 3).

## Environmental sample metabarcoding

Fig 5 shows the results of Oxford Nanopore sequencing on 24 enriched environmental cultures from a dairy farm and the bordering forest. The two species known to be endemic in New Zealand i.e. *L. interrogans* and *L. borgpetersenii*, were present in both the farm and the forest environment. While all samples had *L. borgpetersenii*, only 50% of the samples had *L. interrogans*. A total of 17 of 24 samples had sequences that aligned with *L. borgpetersenii* strain *Pacifica*. In addition, 3 farm samples (SN53, SN65 and SN68) and 1 forest sample (SN79) had sequences that aligned with *Leptospira mayottensis*, 2 forest samples (SN79 and SN81) had sequences that aligned with *Leptopsira alstonii* and *Leptospira tipperaryensis* and 1 farm sample (SN68) and 1 forest sample (SN81) had sequences that aligned with *Leptospira dzianensis*. Forest sample SN78 had sequences that aligned with many species, however, since we do not have isolate controls for these species, we cannot be certain of the level of misclassification between these species given the MinION sequencing error. The proportion of sequences identified as coming from species other than *L. borgpetersenii* and *L. interrogans* ranged from 0.29% to 1.02% in farm samples, and from 0.48% to 74.49% in forest samples.

## Discussion

This study was carried out with the aim of developing a molecular method that can be used for detecting and identifying pathogenic *Leptospira* genomospecies and serovars circulating in New Zealand, with the potential to identify specific strains of interest in human, animal and environmental samples. A One Health approach to pathogen detection and surveillance is essential to develop evidence-based infection control and prevention strategies, particularly for bacteria with complex population and reservoir structures such as *Leptospira*. Serosurveillance has demonstrated that vaccine use is an effective measure to reduce infection in animals and

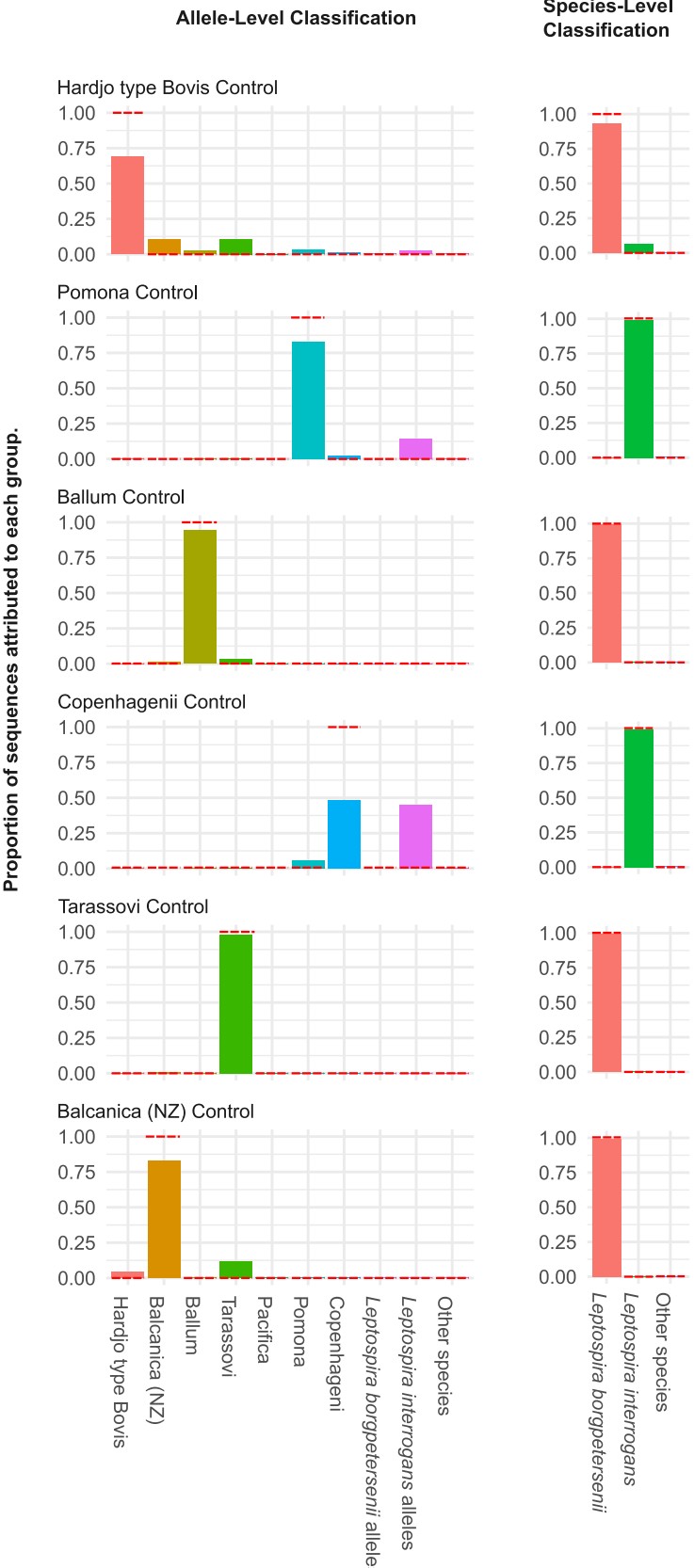

**Fig 2. *Leptospira* variant identification using metabarcoded primers amplifying the *glmU* locus from individual serovar reference-strains using Oxford Nanopore sequencing.** The proportion of reads most closely matching each taxon is indicated at both the allele level (left) and species level (right). Red dashed lines on each bar represent the expected proportion of sequences for each taxon.

humans in New Zealand [17]. Due to the low number of *Leptospira* serovars circulating in New Zealand, detection should be relatively straightforward in people and animals. However, current serological assays in New Zealand only test 8 serovars and current molecular detection does not identify the genomospecies that are associated with these serovars. Furthermore, pathogen identification from the environment is invariably difficult because of the diversity of genomospecies present in any given sample. Thus, environmental detection requires a good understanding of this diversity to establish an effective method to discriminate genomospecies. Because there are many different species of *Leptospira*, highly conserved genes such 16S rRNA are ineffective at discriminating *Leptospira* at species level. Similarly, gene targets specific for *Leptospira* such as *lipL32* also have a low discriminatory power at species level [16]. To identify a locus with enough nucleotide divergence that could be used for strain-level discrimination, we downloaded all pathogenic *Leptospira* genomes published to date, a total of 598 genomes from 20 subclade P1 species and identified the *glmU* locus as having enough nucleotide divergence to discriminate species and serovars individually and in communities in New Zealand (Figs 2 and 4).

Application of this method to enriched environmental cultures identified *L. borgpetersenii* to be more prevalent (24/24 samples tested) than *L. interrogans* (13/24 samples tested) in this farm-forest setting (Fig 5). Furthermore, this study identified *L. borgpeterseneii* strain *Pacifica* in the environment, a strain that to date has been identified in dairy cattle (Fig 4). Lastly, high-identity sequence matches were observed to pathogenic *Leptospira* genomospecies that have not previously been reported in New Zealand. This includes *L. mayottensis* which has previously only been identified from humans in Mayotte, isolated from blood of leptospirosis patients [33]. The 2 *L. mayottensis* isolates identified in Mayotte varied serologically, one belonged to serogroup Mini and the other to serogroup Ballum serovar Kenya [33]. Other species identified from the environmental culture enrichments in this study include *L. tipperaryensis*, *L. dzianensis* and *L. alstonii*. Previous reports of *L. tipperaryensis* include isolation from *Crocidura russula*, the greater white-toothed shrew in Ireland while *L. dzianensis* was isolated from environmental samples in Mayotte and Malaysia [5]. Neither of these species have been characterised serologically. *L. alstonii* has been identified from frogs in China and from environmental samples in Japan and Malaysia [34]. The isolates from China are from serovars Pinchang and Sichuan while the other isolates have not been characterised serologically. To date, *L. tipperaryensis*, *L. dzianensis* and *L. alstonii.* have not been found in humans. Pure culture isolation and further characterisation is required to determine the serogroups and serovars for these strains and species in New Zealand.

While the presented assay can be used in One Health approaches for leptospirosis diagnosis, research and source attribution studies in New Zealand, like most assays for *Leptospira*, the presented methodology has its limitations. The use of this locus for discrimination of serovars other than those present in New Zealand will require validation and will not be directly applicable to all serovars. For example, this method will not be effective for countries that have both *L. interrogans* svs. Copenhageni and Hardjo type Prajitno in circulation as they have identical *glmU* loci. Other conserved domains that span longer genomic regions than the targeted 523 bp of the *glmU* locus may provide better discriminative potential between genomospecies in areas with high *Leptospira* diversity. For example, the 2138 bp *ppk* locus was recently identified as the single core gene of *Leptospira* that provides the most accurate phylogenetic

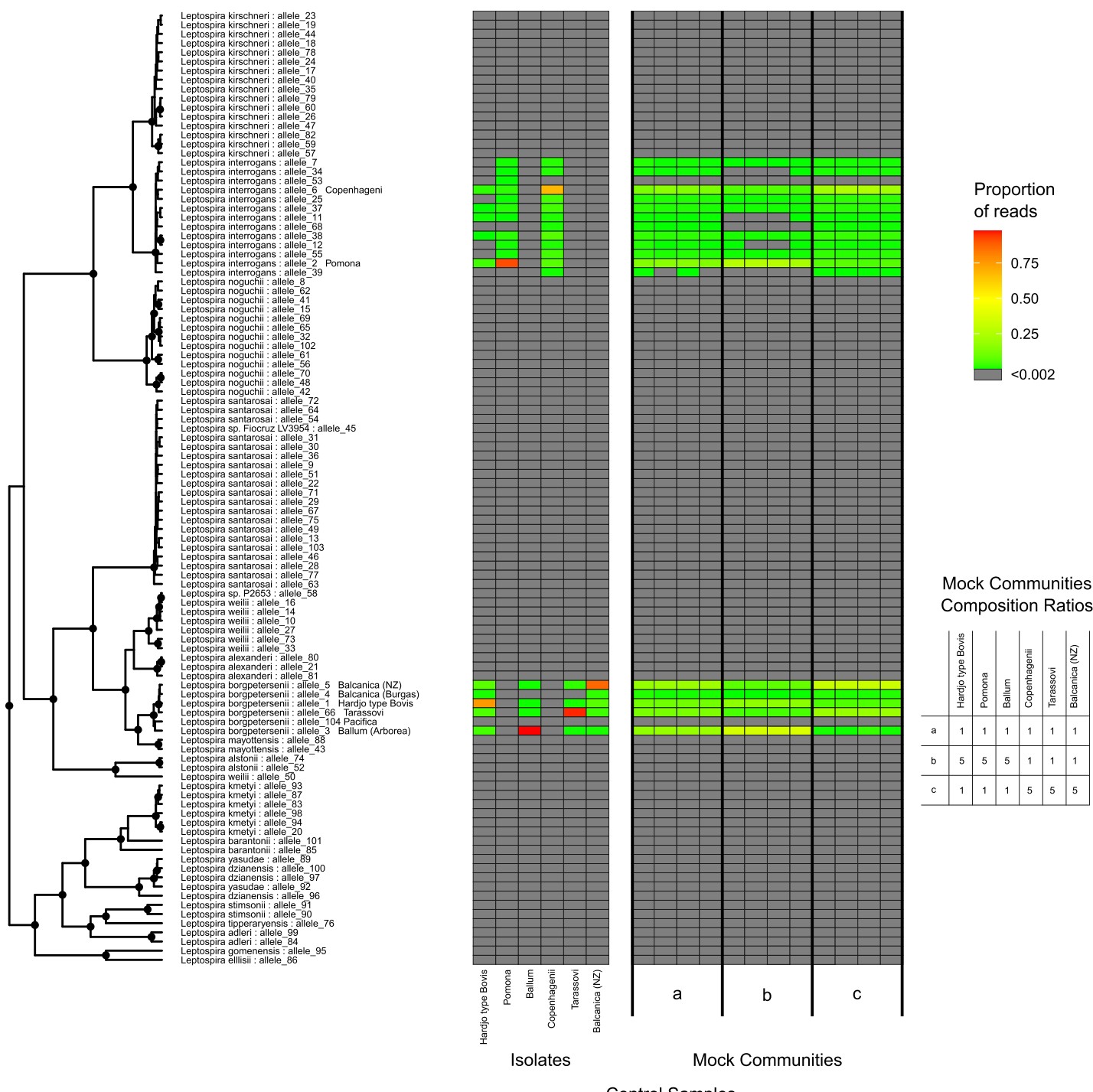

**Fig 3.** *Leptospira* variant identification using metabarcoded primers amplifying the *glmU* locus from individual serovar reference-strains (isolates) and mock communities (a, b, and c) using Oxford Nanopore sequencing. The ratio of DNA in each mock community is presented in the table (bottom right). Results are displayed relative to the phylogenetic position for each allele in the *glmU*-derived phylogenetic tree from Fig 1.

reconstruction of the genus [5]. In addition, for any chosen locus there may be strain variation between isolates from different countries thus, multiple alleles can represent one species and serovar e.g. alleles 4 and 5 have a single nucleotide polymorphism (SNP) but they both

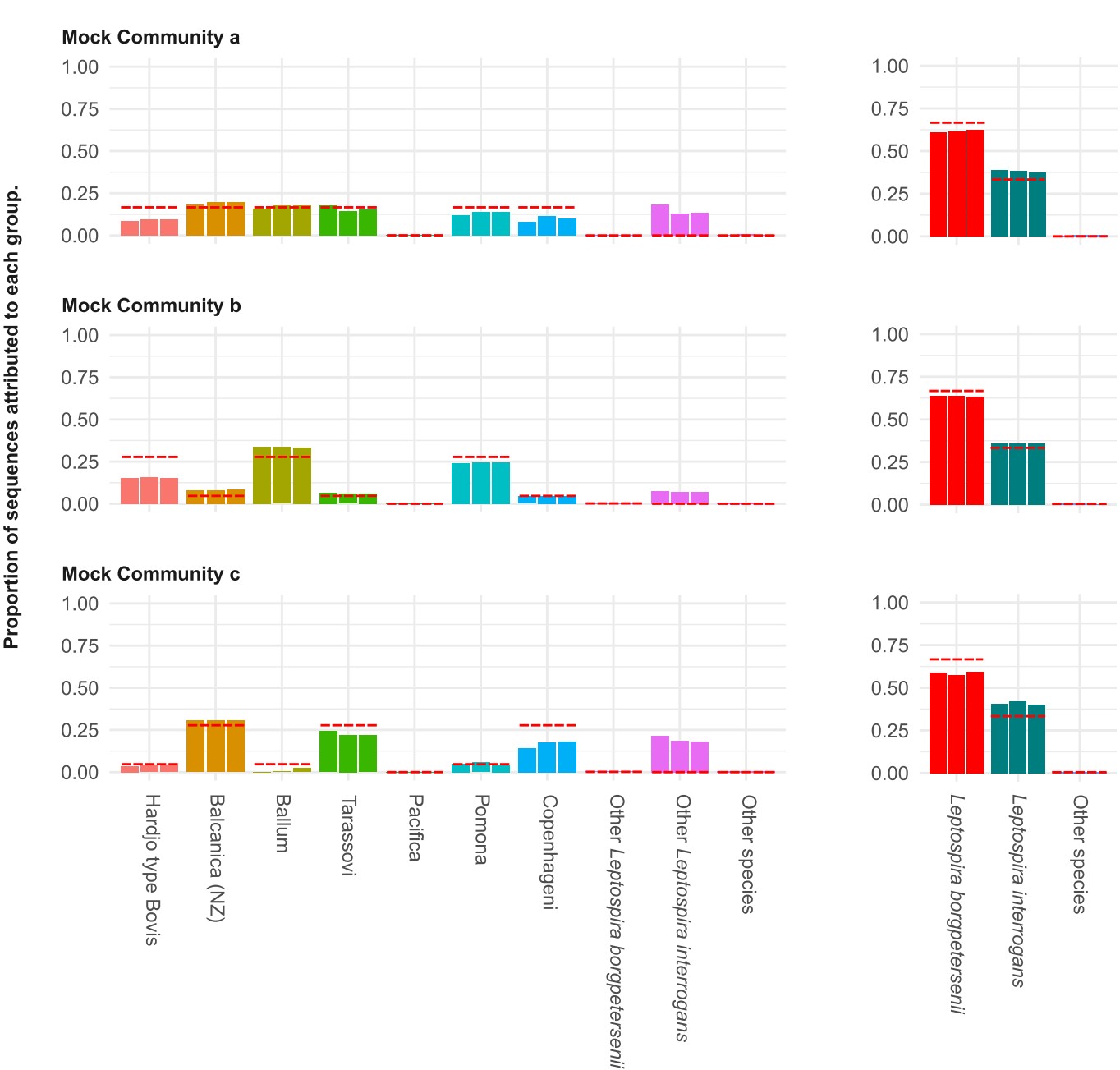

**Fig 4. *Leptospira* variant identification using metabarcoded primers amplifying the *glmU* locus from three mock community samples using Oxford Nanopore sequencing.** Mock community "a" contains a DNA ratio of 1:1 for all serovars, mock community "b" contains a DNA ratio of 5:1 for Hardjo type Bovis, Ballum, Pomona: Balcanica (NZ), Tarassovi, Copenhageni and, mock community "c" contains a DNA ratio of 1:5 for Hardjo type Bovis, Ballum, Pomona: Balcanica (NZ), Tarassovi, Copenhageni. Red dashed lines on each taxon represent the expected proportion of sequences based on the composition of each mock community. Each experiment was performed in triplicate using independently mixed mock communities each time.

represent *L. borgpetersenii* sv. Balcanica, one endemic to New Zealand and the other found overseas. Furthermore, while the application of this method is good for disease surveillance and detection, it does not quantify the full biodiversity of *Leptospira* species i.e. the low-virulence subclade P2 or the saprophytic species from subclades S1 and S2 are not detected as the

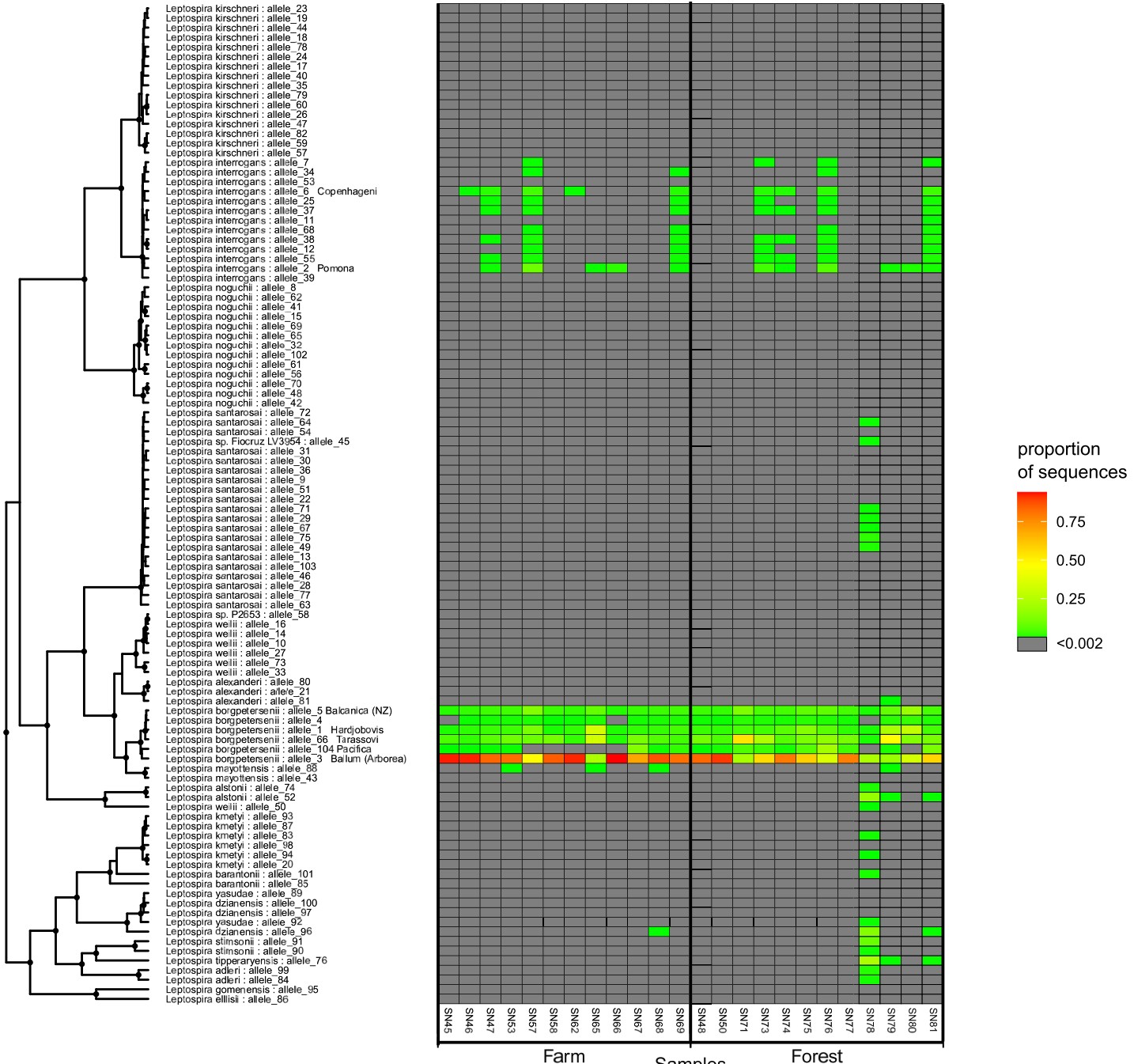

**Fig 5.** *Leptospira* **variant identification using metabarcoded primers amplifying the *glmU* locus from enriched environmental cultures from a dairy farm and a neighbouring forest using Oxford Nanopore sequencing.** Results are displayed relative to the phylogenetic position for each allele in the *glmU*-derived phylogenetic tree.

*glmU* locus is only present in the virulent subclade P1. Lastly, high fidelity sequencing technologies such as Illumina MiSeq may be preferable for short sequence barcoding studies in general, as single nucleotide polymorphisms are used to differentiate serovars and the inherent reduced read accuracy of the Oxford Nanopore sequences leads to observable levels of allele misassignment.

In conclusion, the use of a single genetic locus that allows sensitive and specific detection and whose sequence permits lineage typing is an extremely powerful tool in leptospirosis surveillance. This tool may be particularly applicable in the context of New Zealand where there is a low diversity of known *Leptospira* lineages associated with human disease. This method can be used on both culture and culture-independent samples to increase diagnostic sensitivity, offering new capabilities in leptospirosis research that is valuable across human, animal, and environmental studies. We hope that the application of this method will help to eliminate some of the missing information that persists in the multi-sectoral surveillance and control of leptospirosis.

## Supporting information

**S1 Fig.** Conventional PCR with glmU MLST_2 primers (top) and *glmU*_DW primers (bottom). LD = 1kb+ ladder, NC = negative control, PC = positive control (*L. borgpetersenii* serovar Hardjo DNA), lanes 1–10 = PCR products amplified from gDNA extracted from cattle urine. Samples 6, 7, 9 and 10 from the bottom gel were verified as Leptospira with Sanger sequencing.
(DOCX)

**S1 Table. Bacterial strains used for metabarcoding controls.**
(DOCX)

**S2 Table. Primers analysed or used in this study.**
(DOCX)

## Acknowledgments

We would like to thank Neville Haack, Emilie Vallee, Cord Heuer, Peter Wilson, Julie Collins-Emerson, Yupi Yupiana, Fang Fang, Marie Moinet, Mark van de Pol, Sithar Dorjee and Supatsak Subharat for their involvement in projects that led to the collection of isolates for whole genome sequence analysis; Marga Goris for providing serovar Arborea DNA and for testing *L. borgpetersenii* serovars Ballum and Arborea cross-reactivity (Leptospira Reference Centre, Academic Medical Centre, Netherlands) and Anne C. Midwinter for her general support, and for setting up the Oxford Nanopore sequencing platform, the MinION at mEpiLab.

## Author Contributions

**Conceptualization:** David A. Wilkinson, Jackie Benschop, Shahista Nisa.

**Data curation:** David A. Wilkinson, Shahista Nisa.

**Formal analysis:** David A. Wilkinson, Matthew Edwards, Shahista Nisa.

**Funding acquisition:** Shahista Nisa.

**Investigation:** David A. Wilkinson, Matthew Edwards, Shahista Nisa.

**Methodology:** David A. Wilkinson, Matthew Edwards, Shahista Nisa.

**Supervision:** Jackie Benschop.

**Validation:** David A. Wilkinson.

**Visualization:** David A. Wilkinson.

**Writing – original draft:** David A. Wilkinson, Jackie Benschop, Shahista Nisa.

**Writing – review & editing:** David A. Wilkinson, Matthew Edwards, Jackie Benschop, Shahista Nisa.

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
