## [Decision Letter · Decision Letter 0]

7 Jun 2021

PONE-D-21-10197

Identification of pathogenic Leptospira species and serovars in New Zealand using metabarcoding.

PLOS ONE

Dear Dr. Wilkinson,

Thank you for submitting your manuscript to PLOS ONE. After careful consideration, we feel that it has merit but does not fully meet PLOS ONE’s publication criteria as it currently stands. Therefore, we invite you to submit a revised version of the manuscript that addresses the points raised during the review process.

Appropriate negative control is emphasized to validate the applied technology for the species level identification of Leptospira. Quality of the figures to be improvised.

We look forward to receiving your revised manuscript.

Kind regards,

Kalimuthusamy Natarajaseenivasan

Academic Editor

PLOS ONE

Journal Requirements:

2. In the interests of reproducibility, in your Methods, please provide the name and location of the farm from which samples were obtained from your study. If you have been advised by an institutional data protection officer or equivalent authority that the identity of the farm should remain anonymous, please update your Data Availability Statement to mention this.

3.Thank you for stating the following in the Funding Section of your manuscript:

"This project was funded by the Hawke’s Bay Medical Research Foundation and the Palmerston

386 North Medical Research Foundation. Our research into leptospirosis in New Zealand is

387 additionally supported by the Health Research Council."

Reviewers' comments:

Reviewer's Responses to Questions

**Comments to the Author**

1. Is the manuscript technically sound, and do the data support the conclusions?

Reviewer #1: Yes

Reviewer #2: Partly

Reviewer #3: Partly

2. Has the statistical analysis been performed appropriately and rigorously? 

Reviewer #1: Yes

Reviewer #2: N/A

Reviewer #3: N/A

3. Have the authors made all data underlying the findings in their manuscript fully available?

Reviewer #1: Yes

Reviewer #2: Yes

Reviewer #3: No

4. Is the manuscript presented in an intelligible fashion and written in standard English?

Reviewer #1: Yes

Reviewer #2: Yes

Reviewer #3: Yes

5. Review Comments to the Author

Reviewer #1: Overall, the manuscript is well-written. However, the manuscript could be further strengthened in describing the gap of knowledge or research problem in a clear manner and stating the challenges faced during trials prior to obtaining the results.

Reviewer #2: This Research Article by Wilkinson et al, described the introduction of a new methodology to identify pathogenic Leptospira species in enriched environmental cultures. The authors identified a partial region of the glmU gene as a suitable locus for the discrimation of Leptospira species and endemic serovars of New Zealand and explored the use of this locus using the MinION sequencing platform. Since current molecular methods used for Leptospira detection are unable to discriminate species and serovars, the introduction of this technology in the field of leptospirosis will be a valuable tool for epidemiological surveys and will also increase the understanding of Leptospira diversity in New Zealand allowing the implementation of control strategies of this important zoonotic disease.

Major comments:

1. The Nanopore Sequencing Technology is used for identification of bacterial communities in the field and provides real-time data enabling immediate access to results such as species identification and abundance (Urban, Holzer, et al. eLife 2021;10:e61504. DOI: https://doi.org/10.7554/eLife.61504). My major concern for this study is about the enrichment of the environmental samples by culturing in EMJH broth for 13 weeks. One important information is to know which species of Leptospira are present in the environment and at which levels. The direct sequencing of DNA from environmental samples overcomes enrichment and biases common to culturing (i.e.: some Leptospira species growth faster than others). Did the authors try to sequence glmU by using the DNA extracted from the environmental water samples (without culture enrichment)?

2. This study should include appropriate negative controls such as bacterial strains that are not expected to be detected with the designed primers (i.e. Leptospira subclade S1 and/or other environmental bacterial species) and create mixed mock communities including DNA preparations from negative controls.

3. A shorter incubation time than 13 weeks to enrich cultures will be practical from the clinical point of view if this method is going to be use for diagnosis. What is the rationale for incubating EMJH cultures 13 weeks? Detectable leptospiral DNA might be present in the enriched culture before leptospires are visible by dark field microscopy. Did the author test DNA extracted from younger cultures?

4. The taxonomy of the genus Leptospira was revisited by Vincent et al. (2019) and the genus was divided into 2 clades and 4 subclades (P1, P2, S1 and S2). Subclade P1 can further split into “virulent” and “low-virulence” pathogens defined by distinct accessory gene patterns and ability to cause or not severe infections in animals and humans (Thibeaux et al, https://doi.org/10.1099/mgen.0.000144). Virulent pathogens species (L. interrogans, L. borgpetersenii, L. kershneri, among others) are the main agents of leptospirosis, while the virulence in humans for some “low-virulence” pathogens species is still controversial. Thus, I suggest to update the classification of the genus Leptospira in Introduction (lines 50-51), and describe the Results with the proper Discussion considering the concept of “low-virulence”.

Minor comments:

1. Abstract, line 38: correct “environmental samples” to “enriched environmental cultures”. The detection of Leptospira species from environmental water samples have been described (Gamage et al., doi: 10.1371/journal.pntd.0008437; Urban, Holzer, et al. eLife 2021;10:e61504. DOI: https://doi.org/10.7554/eLife.61504); thus, the proper terminology throughout the MS is recommended.

2. Abstract, line 30-31: what is the meaning of “reported cases that have not been attributed to the infecting serovar or genomospecies”? These are clinically-diagnosed cases but not laboratory-confirmed cases? Please explain.

3. Introduction, line 80: “pathogenic species”. Please specify which ones or described to which subclade they belong to.

4. Introduction, line 96. Please give a reference for “different reservoir distributions for different serovars”.

5. Introduction, line 101: correct the spelling for “Leptospira”.

6. Discussion (lines 334-344). Discuss Leptospira species and serovars by explaining the virulence of each Leptospira species in humans.

7. Discussion (lines 352-355). Vincent et al. (Reference No. 36 of the current MS) reported that the ppk gene can be used to discriminate Leptospira genomospecies. Did the authors consider this gene? Please discuss about this locus.

8. Discussion lines 368-369: “in the context of New Zealand where low genomospecies and serovar diversity”... ? Since one of the conclusion of this study is that several species are present in environmental water samples of New Zealand , and Leptospira species diversity seems to be higher than expected, the phrase “low genomospecies diversity” is contradictory. Please re-phase.

Reviewer #3: The authors tried to detect pathogenic Leptospira DNA from environmental samples of New Zealand. They focused on long read DNA sequencing technology using Oxford Nanopore MinION to distinguish intra-species variations of pathogenic Leptospira.

I think the aim of this study is very nice and preferable. The prior identification of a suitable marker gene glmU based on genome sequence analysis of Leptospira species is very valuable.

In my opinion, however, unfortunately, the Figures of the present manuscript seems to be low quality.

(1) Figure 1: OTU names is invisible and unreadable.

(2) Figure 2: the correspondence between bar plot and the caption is very obscure. The green bar and the caption "Tarassovi" (?) seem to be already misaligned. The correct caption of the blue and magenta bars on the right side is not identifiable.

(3) Figure 3: OTU names, names of the "Isolates", and values in the "Mock Communities Composition Ratios" are unreadable.

(4) Figure 5: OTU names and captions below the heatmap panel are unreadable.

The legends of the figures does not cover these insufficiencies of the figures above.

Accordingly, in my view, the readers are not able to evaluate the validity of the results and discussion based on these figures. I think the manuscript is better to be rejected once, and recommended to resubmit after correction of the figures and the related things.

Minor points:

Line 173: the version of the MEGA X used should be described.

Line 221: "amplified" would be a possible typo of "amplify."

Line 222: "LipL32" may be expressed as italicized "lipL32."

Line 296: "of24" seems to miss the space.

Line 321: I think the s of the "16s rRNA" is usually described in uppercase. (16S rRNA; Sedimentation coefficient)

Line 323: "LipL32" may be expressed as italicized "lipL32."

Line 483: "... controls, results are ..." may be a typo of "... controls. Results are ...".

6. PLOS authors have the option to publish the peer review history of their article (what does this mean?). If published, this will include your full peer review and any attached files.

Reviewer #1: **Yes: **Lesley Maurice Bilung

Reviewer #2: No

Reviewer #3: No

---

## [Author Response · Author response to Decision Letter 0]

22 Jul 2021

Dear PLOS Editor,

We would like to thank you and the three reviewers of our manuscript for taking the time to appraise our work. We have been through the comments carefully and edited our manuscript accordingly. We believe that the modifications have significantly improved the clarity of the article, and we hope that the revised version of our paper can now be considered for publication in PLOS ONE.

We provide point-by-point responses below.

Reviewers’ comments are in red italic, our responses are in black.

Yours Sincerely,

Dr David A Wilkinson

 

EDITORIAL COMMENTS:

We have been through the article and modified the document formatting in accordance with the PLOS ONE template. Many thanks for these guidelines. 

2. In the interests of reproducibility, in your Methods, please provide the name and location of the farm from which samples were obtained from your study. If you have been advised by an institutional data protection officer or equivalent authority that the identity of the farm should remain anonymous, please update your Data Availability Statement to mention this.

The owners of the farm would like its exact location to remain anonymous. We have added a statement to the Data Availability section to this effect.

3.Thank you for stating the following in the Funding Section of your manuscript:

"This project was funded by the Hawke’s Bay Medical Research Foundation and the Palmerston

386 North Medical Research Foundation. Our research into leptospirosis in New Zealand is

387 additionally supported by the Health Research Council."

The funding statement has been added to the revision’s cover letter and removed from the body of the manuscript. 

Done

 

Review Comments to the Author

Reviewer #1: Overall, the manuscript is well-written. However, the manuscript could be further strengthened in describing the gap of knowledge or research problem in a clear manner and stating the challenges faced during trials prior to obtaining the results.

Many thanks for this comment. We have added text to the introduction and discussion sections that make the aim of the study, its context and the issues faced in leptospirosis diagnostics in New Zealand more explicit. We hope that this has clarified those aspects that the reviewer found to be lacking.

 

Reviewer #2: This Research Article by Wilkinson et al, described the introduction of a new methodology to identify pathogenic Leptospira species in enriched environmental cultures. The authors identified a partial region of the glmU gene as a suitable locus for the discrimation of Leptospira species and endemic serovars of New Zealand and explored the use of this locus using the MinION sequencing platform. Since current molecular methods used for Leptospira detection are unable to discriminate species and serovars, the introduction of this technology in the field of leptospirosis will be a valuable tool for epidemiological surveys and will also increase the understanding of Leptospira diversity in New Zealand allowing the implementation of control strategies of this important zoonotic disease.

Many thanks for this positive assessment. We also believe that these sorts of barcoding approaches will be of great use in leptospirosis diagnostics (as highlighted recently in a paper by Guernier et al.).

Major comments:

1. The Nanopore Sequencing Technology is used for identification of bacterial communities in the field and provides real-time data enabling immediate access to results such as species identification and abundance (Urban, Holzer, et al. eLife 2021;10:e61504. DOI: https://doi.org/10.7554/eLife.61504). My major concern for this study is about the enrichment of the environmental samples by culturing in EMJH broth for 13 weeks. One important information is to know which species of Leptospira are present in the environment and at which levels. The direct sequencing of DNA from environmental samples overcomes enrichment and biases common to culturing (i.e.: some Leptospira species growth faster than others). Did the authors try to sequence glmU by using the DNA extracted from the environmental water samples (without culture enrichment)?

We agree with the reviewer that it is not ideal to rely on 13-week enrichment cultures for Leptospira sequencing. This is, indeed, one of the main reasons for using barcoding techniques – to bypass the need for culture-dependent methods. However, the specific samples that we tested in this study are the perfect example of why barcoding techniques can be useful when culture-dependent techniques fail. We had attempted to isolate clonal cultures from these samples for a long period of time, specifically Pacifica, without success. Sanger sequencing of PCR products gave mixed signals, and there was high cellular heterogeneity in dark field observations from these cultures. It was only by developing this barcoding technique that we were able to identify what was in these mixed cultures (without resorting to cloning PCR products). We agree that the enrichment culture is likely to have introduced biases in the distributions of different lineages in each sample, however, determining the distributions of species of Leptospira in the environment was not the primary purpose of this body of work. For this study, we were exploring the use of thie glmU locus as a molecular barcoding tool via the Oxford Nanopore Technology (ONT) sequencing platform MinION i.e. method validation rather than researching the environment.

The technique itself will work on any sample from which a PCR product can be obtained. Our estimates of the limit of detection for our PCR primers suggest that this would even be true from very dilute cultures – and many environmental samples prior to enrichment.

2. This study should include appropriate negative controls such as bacterial strains that are not expected to be detected with the designed primers (i.e. Leptospira subclade S1 and/or other environmental bacterial species) and create mixed mock communities including DNA preparations from negative controls.

As stated in the manuscript, negative controls were included in all limit-of-detection and mixed mock community PCR amplifications. In addition, we have additional data from animal urine and kidneys that suggests there is little or no “false positive” amplification from samples that have tested negative using alternative PCR systems such as the lipL32 system.

It was not (and is not) possible for us to include Leptospira controls from other subclades, as we have no access to these strains and the rules preventing import of potential zoonotic pathogens to New Zealand are very strict.

3. A shorter incubation time than 13 weeks to enrich cultures will be practical from the clinical point of view if this method is going to be use for diagnosis. What is the rationale for incubating EMJH cultures 13 weeks? Detectable leptospiral DNA might be present in the enriched culture before leptospires are visible by dark field microscopy. Did the author test DNA extracted from younger cultures?

As stated in major comment 1 above, the described method will work for any sample from which a PCR product can be obtained. The limit of detection of the PCR system was calculated, meaning we know that PCR products of sequencing quality can be obtained from samples that have concentrations as low as 10 leptospires per ml. 

4. The taxonomy of the genus Leptospira was revisited by Vincent et al. (2019) and the genus was divided into 2 clades and 4 subclades (P1, P2, S1 and S2). Subclade P1 can further split into “virulent” and “low-virulence” pathogens defined by distinct accessory gene patterns and ability to cause or not severe infections in animals and humans (Thibeaux et al, https://doi.org/10.1099/mgen.0.000144). Virulent pathogens species (L. interrogans, L. borgpetersenii, L. kershneri, among others) are the main agents of leptospirosis, while the virulence in humans for some “low-virulence” pathogens species is still controversial. Thus, I suggest to update the classification of the genus Leptospira in Introduction (lines 50-51), and describe the Results with the proper Discussion considering the concept of “low-virulence”.

This has been updated as requested, and we now refer to relevant subclades P1, P2, S1 and S2 throughout the manuscript. Specifically, lines 39-45 in the introduction and lines 355-356 in the discussion.

Minor comments:

1. Abstract, line 38: correct “environmental samples” to “enriched environmental cultures”. The detection of Leptospira species from environmental water samples have been described (Gamage et al., doi: 10.1371/journal.pntd.0008437; Urban, Holzer, et al. eLife 2021;10:e61504. DOI: https://doi.org/10.7554/eLife.61504); thus, the proper terminology throughout the MS is recommended.

We have corrected the way in which we refer to these samples and used “enriched environmental cultures” throughout.

2. Abstract, line 30-31: what is the meaning of “reported cases that have not been attributed to the infecting serovar or genomospecies”? These are clinically-diagnosed cases but not laboratory-confirmed cases? Please explain.

This refers to individual cases that have been diagnosed as positive for leptospirosis through PCR – only positive/negative data is available, but no typing information is available… for example, in contrast to what would be obtained when using MAT which would give serological results against one or several specific serovars.

We have slightly altered the wording of this sentence to clarify the message.

3. Introduction, line 80: “pathogenic species”. Please specify which ones or described to which subclade they belong to.

We have edited this and are referring to this as the P1 species throughout the paper.

4. Introduction, line 96. Please give a reference for “different reservoir distributions for different serovars”.

A suitable reference has been added to the text.

5. Introduction, line 101: correct the spelling for “Leptospira”.

Done.

6. Discussion (lines 334-344). Discuss Leptospira species and serovars by explaining the virulence of each Leptospira species in humans.

Thank you for raising this. Additional discussion has been added to lines 333-344.

7. Discussion (lines 352-355). Vincent et al. (Reference No. 36 of the current MS) reported that the ppk gene can be used to discriminate Leptospira genomospecies. Did the authors consider this gene? Please discuss about this locus.

This is a very good point. The ppk locus is indeed a good choice for the discrimination of different Leptospira. Vincent et al. report that it is a core gene that allows the accurate reconstruction of the genome-level phylogeny. Indeed, when we investigated the discriminatory power of this locus for New Zealand strains, it does allow the identification of all known endemic human pathogenic Leptospira. However, we did not consider this locus because no conserved primer scheme has been proposed for its amplification to date (to our knowledge). Following the reviewer’s comment, we examined the locus in more detail. Upon alignment of ppk sequences from all available Leptospira genomes, we were unable to identify conserved flanking sequences that would allow for its effective use. It is possible that the use of degenerate primers may allow its amplification however, we specifically avoided the use of degenerate primers in our study – as there may exist differences in sensitivity for the different primer sequences.

We have added a sentence to the discussion of the article specifically mentioning ppk as a potential locus for further investigation of metabarcoding protocols.

8. Discussion lines 368-369: “in the context of New Zealand where low genomospecies and serovar diversity”... ? Since one of the conclusion of this study is that several species are present in environmental water samples of New Zealand , and Leptospira species diversity seems to be higher than expected, the phrase “low genomospecies diversity” is contradictory. Please re-phase.

This is a good point. When we refer to low Leptospira diversity, specifically we are talking about the diversity of known species associated with human disease. The phrase has been altered accordingly. 

Reviewer #3: The authors tried to detect pathogenic Leptospira DNA from environmental samples of New Zealand. They focused on long read DNA sequencing technology using Oxford Nanopore MinION to distinguish intra-species variations of pathogenic Leptospira.

I think the aim of this study is very nice and preferable. The prior identification of a suitable marker gene glmU based on genome sequence analysis of Leptospira species is very valuable.

Many thanks for these positive comments.

In my opinion, however, unfortunately, the Figures of the present manuscript seems to be low quality.

(1) Figure 1: OTU names is invisible and unreadable.

(2) Figure 2: the correspondence between bar plot and the caption is very obscure. The green bar and the caption "Tarassovi" (?) seem to be already misaligned. The correct caption of the blue and magenta bars on the right side is not identifiable.

(3) Figure 3: OTU names, names of the "Isolates", and values in the "Mock Communities Composition Ratios" are unreadable.

(4) Figure 5: OTU names and captions below the heatmap panel are unreadable.

The legends of the figures does not cover these insufficiencies of the figures above.

Accordingly, in my view, the readers are not able to evaluate the validity of the results and discussion based on these figures. I think the manuscript is better to be rejected once, and recommended to resubmit after correction of the figures and the related things.

I am sorry that you were unable to interpret the figures for the manuscript. A major reason for this may be due to the revision system, which can often generate low quality figures due to the pdf conversion process. Given that other reviewers were able to read and interpret the figures, I believe this to be a conversion issue and not an actual issue with the figures. However, in an attempt to make the figures more legible, we have increased font sizes where possible and corrected any errors that have been pointed out. We have also submitted vector format .eps files for each figure, which should allow PLOS to provide a high-resolution figure when in print. Hopefully this will make the figures easier to interpret.

Minor points:

Line 173: the version of the MEGA X used should be described.

The version number has been added to the text.

Line 221: "amplified" would be a possible typo of "amplify."

Well spotted. This has been changed.

Line 222: "LipL32" may be expressed as italicized "lipL32."

We have italicized lipL32 whenever it has been used to refer to a specific genetic locus in the text.

Line 296: "of24" seems to miss the space.

Changed.

Line 321: I think the s of the "16s rRNA" is usually described in uppercase. (16S rRNA; Sedimentation coefficient)

Absolutely correct. We have changed the text.

Line 323: "LipL32" may be expressed as italicized "lipL32."

We have italicized lipL32 whenever it has been used to refer to a specific genetic locus in the text.

Line 483: "... controls, results are ..." may be a typo of "... controls. Results are ...".

Changed.

---

## [Decision Letter · Decision Letter 1]

15 Sep 2021

Identification of pathogenic Leptospira species and serovars in New Zealand using metabarcoding.

PONE-D-21-10197R1

Dear Dr. Wilkinson,

We’re pleased to inform you that your manuscript has been judged scientifically suitable for publication and will be formally accepted for publication once it meets all outstanding technical requirements.

Kind regards,

Kalimuthusamy Natarajaseenivasan

Academic Editor

PLOS ONE

Additional Editor Comments (optional):

Reviewers' comments:

Reviewer's Responses to Questions

**Comments to the Author**

1. If the authors have adequately addressed your comments raised in a previous round of review and you feel that this manuscript is now acceptable for publication, you may indicate that here to bypass the “Comments to the Author” section, enter your conflict of interest statement in the “Confidential to Editor” section, and submit your "Accept" recommendation.

Reviewer #1: All comments have been addressed

Reviewer #2: All comments have been addressed

Reviewer #3: (No Response)

2. Is the manuscript technically sound, and do the data support the conclusions?

Reviewer #1: Yes

Reviewer #2: (No Response)

Reviewer #3: Yes

3. Has the statistical analysis been performed appropriately and rigorously? 

Reviewer #1: Yes

Reviewer #2: (No Response)

Reviewer #3: N/A

4. Have the authors made all data underlying the findings in their manuscript fully available?

Reviewer #1: Yes

Reviewer #2: (No Response)

Reviewer #3: Yes

5. Is the manuscript presented in an intelligible fashion and written in standard English?

Reviewer #1: Yes

Reviewer #2: (No Response)

Reviewer #3: Yes

6. Review Comments to the Author

Reviewer #1: (No Response)

Reviewer #2: (No Response)

Reviewer #3: I think that the authors adequately addressed the concerns from the reviewers regarding previous version of the manuscript. It seems that the revised version of the manuscript is now acceptable.

Only the following minor points, I hope that authors will confirm. I think this potential correction could be done after acceptance of the paper, at the stage of proof production:

(1) The digit expression, please re-check and unify:

Line 120: at 8 000 x ...

Line 134: from 200 000 to ...

Line 191: 1500 fmol

Line 237: of 1:10240 while ...

Line 238: of 1:1280 ...

Line 357: the 2138 bp ppk ...

7. PLOS authors have the option to publish the peer review history of their article (what does this mean?). If published, this will include your full peer review and any attached files.

Reviewer #1: No

Reviewer #2: No

Reviewer #3: No

---

## [Editor Report · Acceptance letter]

20 Sep 2021

PONE-D-21-10197R1 

Identification of pathogenic *Leptospira* species and serovars in New Zealand using metabarcoding. 

Dear Dr. Wilkinson:

I'm pleased to inform you that your manuscript has been deemed suitable for publication in PLOS ONE. Congratulations! Your manuscript is now with our production department. 

Kind regards, 

on behalf of

Dr. Kalimuthusamy Natarajaseenivasan 

Academic Editor

PLOS ONE